# Exploring Multi-hop Reasoning Process in NLU from the View of Bayesian Probability

**Yitian Li**[1,2] , **Jidong Tian**[1,2] , **Hao He**[1,2*] and **Yaohui Jin**[1,2*]

[1]MoE Key Lab of Artificial Intelligence, AI Institute, Shanghai Jiao Tong University

[2]State Key Lab of Advanced Optical Communication System and Network, Shanghai Jiao Tong University

{ yitian_li, frank92, hehao, jinyh}@sjtu.edu.cn

## Abstract

Emerging pre-trained language models (PTLMs), such as BERT and RoBERTa, have already achieved great success on many natural language understanding (NLU) tasks, spurring widespread interest for their potential in scientific and social areas, with accompanying criticism on their ambiguousness of reasoning, especially multi-hop cases. Concretely, many studies have pointed out that these models lack true understandings of the reasoning process. In this work, we focus on multi-hop reasoning processes of PTLMs and perform an analysis on a logical reasoning dataset, Soft Reasoner. We first extend the dataset by constructing the implicit intermediate results during multi-hop reasoning in a semi-automatic way. Surprisingly, when testing on the extended dataset, PTLMs can even predict the correct conclusion when they cannot judge the corresponding intermediate results. To further analyze this phenomenon, we further compare PTLMs' reasoning processes with Bayesian inference processes to simulate humans' reasoning procedure. Results show that if a model is more in line with the Bayesian process, it tends to have a better generalization ability. Our Bayesian process method can be used as a method to evaluate the generalization ability of models.

## 1 Introduction

Natural language understanding (NLU) is one of the critical problems in NLP, which aims to empower machines to understand language generated by humans [Hossain *et al.*, 2020]. Many NLU systems, especially Transformers-based [Vaswani *et al.*, 2017] language models (BERT [Devlin *et al.*, 2019], RoBERTa [Liu *et al.*, 2019]), seem to be successful on many NLU-related tasks, such as question answering (QA) and natural language inference (NLI) [Sinha *et al.*, 2019]. However, it is still inconclusive whether these models can truly understand natural language and make reasonable decisions or not [Niven and Kao, 2019]. On the one hand, some evidence shows that PTLMs lack sufficient reasoning

---
*Corresponding Authors

ability in complex reasoning tasks [Bhagavatula *et al.*, 2020; Liu *et al.*, 2020]. On the other hand, many studies point out that correct decisions may come from spurious statistical correlations rather than true reasoning abilities, resulting in poor generalization and robustness [Jiang and Bansal, 2019; Kaushik and Lipton, 2018; Gururangan *et al.*, 2018]. Therefore, there has been an increasing need to understand how these PTLMs work [Misra *et al.*, 2020].

In this work, we explore to analyze how PTLMs work in NLU by introducing a novel simulation of the reasoning process. In reality, some classic studies have extracted attention that implicitly reflects the reasoning process [Serrano and Smith, 2019; Abnar and Zuidema, 2020], while others directly provide explicit evidence or proof to evaluate models [Gontier *et al.*, 2020]. Furthermore, a recent kind of method takes advantage of counterfactual instances to perform analysis or attack models, which is another way to understand the limitations of NLU models [Kurakin *et al.*, 2017; Jin *et al.*, 2020]. These studies provide significant views on how to use the reasoning process to understand PTLMs' reasoning ability, but these studies hardly take into account the probabilistic reasoning process of neural models.

Based on the above analysis, there are two key points when describing the reasoning process of PTLMs. Firstly, according to the intuition of the human reasoning process, if a model truly understands the reasoning process, it is also supposed to make correct predictions of the intermediate results. For example, when we have known the premises (*Bob likes adorable animals. Luna is a cat. A cat is an animal. The cat is adorable.*) and wanted to judge the hypothesis (*Bob likes Luna.*), Humans are easy to conclude the intermediate results ( *Luna is an animal. Luna is adorable.*). Secondly, we build the analytical method that could measure the probability distributions of the whole reasoning process rather than only provide deterministic reasoning results.

Therefore, we concentrate on the reasoning process of multi-hop reasoning in NLU, and our proposed analytical method also introduces the explicit intermediate results to describe the reasoning process. Differently, we next introduce the Bayesian network to describe the probabilistic reasoning process of humans based on the intermediate results and compare the PTLMs' reasoning processes with such a network. As the previous work of Wang et al. [Wang *et al.*, 2019] mentions, a neural model inferring through the correct

reasoning process has better generalization ability on zero-shot evaluations. Our experimental results on a logical benchmark dataset, Soft Reasoner, further support this view from the probabilistic perspective, which means that a model better conforming to the Bayesian network is easier to generalize.

Our main contributions are summarized as:

- We propose a novel analytical method to evaluate how PTLMs perform on multi-hop reasoning problems of NLU. This method takes advantage of explicit intermediate results to construct the Bayesian network that benefits to model a human-like and probabilistic reasoning process.

- Experiments on the Soft Reasoner dataset provide evidence that if a model is more in line with the Bayesian process, it seems to make a more human-like reasoning process, making it easier to generalize.

## 2 Related Work

### 2.1 Benchmarking Natural Language Reasoning

Researches on evaluating how PTLMs perform reasoning tasks have emerged quickly [McCoy *et al.*, 2019]. Most of these studies are based on three different focuses of the reasoning process: attention, proof, and counterfactual samples. Attention-based methods [Serrano and Smith, 2019; Abnar and Zuidema, 2020] take advantages of the attention mechanism to provide the posterior validation to explain the reasoning process. Proof-based methods [Gontier *et al.*, 2020] provide the priori reasoning path to evaluate models. Attack-based methods [Kurakin *et al.*, 2017; Jin *et al.*, 2020] construct counterfactual scenarios to explore models' limitations. These studies have brought detailed views on PTLMs. For example, Niven and Kao [Niven and Kao, 2019] found that BERT tends to exploit the presence of cue words to predict, such as "not". There are also studies [Gururangan *et al.*, 2018; Poliak *et al.*, 2018] that exposed how biases in datasets influence PTLMs. Other findings [Glockner *et al.*, 2018; Carmona *et al.*, 2018] also revealed that some NLI models might use fallible heuristics to make decisions. However, these studies are not adequate in analyzing the reasoning process of PTLMs as they neglected the probabilistic characteristics of neural models.

### 2.2 Probabilistic Logical Reasoning

Considering reasoning in NLU, it is reasonable to involve logical methods into neural models [Lage *et al.*, 2018]. Although traditional methods used hard logic rules [Qu and Tang, 2019], more preferable methods for logical reasoning in NLU are probabilistic logical methods that better match the probabilistic characteristics of neural models, such as Markov logic network (MLN) [Richardson and Domingos, 2006; Singla and Domingos, 2005] and DistMult [Yang *et al.*, 2015]. Recently, Manhaeve et al. [Manhaeve *et al.*, 2018] proposed a probabilistic logic programming method that can fit the neural models perfectly. Qu and Tang [Qu and Tang, 2019] proposed a probabilistic framework, pLogicNet, that can use logic rules to handle uncertainty in a principled way based on Markov logic networks. Although these methods

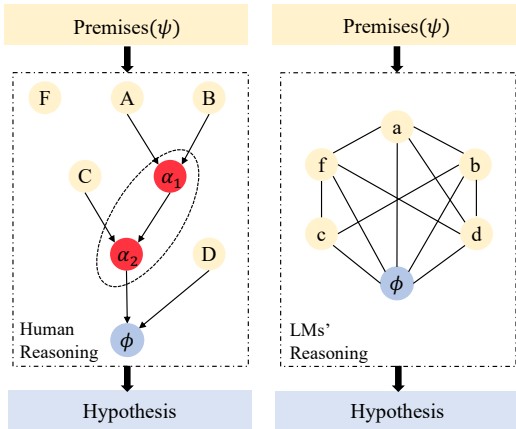

Figure 1: Comparison of the human reasoning process and the LMs' reasoning process. The yellow circles, including $F, A, B, C, D$, are the perceived from premises ($\psi$) directly, and their corresponding lowercase letters indicate their hidden space, while the red-bordered circles ($\alpha_1$, $\alpha_2$) are the intermediate results in the Bayesian Network. The blue circle is the hypothesis ($\phi$).

focus more on integrating probabilistic logical reasoning and neural networks, their effectiveness also supports the view that understanding the reasoning process of neural models requires considering their probabilistic characteristics.

## 3 Methodology

### 3.1 Probabilistic Reasoning Process

In this paper, we introduce the directed acyclic graph (DAG) to model the human reasoning process as it can describe logical dependence among different propositions similar to humans [Chen *et al.*, 2020; Talvitie and Koivisto, 2019]. However, neural networks always make reasoning through a fully-connected bidirectional graph encoded in the neurons. Comparisons are shown in Figure 1. To measure the similarity between the right graph and the left one in Figure 1, we are required to generate the intermediate results $\alpha_1$ and $\alpha_2$, and evaluate the connections in the right graph according to paths defined in DAG.

Besides, PTLMs' reasoning process is a probabilistic reasoning process, especially in NLU [Petroni *et al.*, 2019]. When given the context $x$ and a proposition $z$ to be judged, we use the form of $[CLS]x[SEP]z[SEP]$ as the input of PTLMs. Then, PTLMs will output the probability $P(z|x)$ calculated by the following equation, $h_\theta$ is a scoring function or a negative energy function represented by a neural network with parameters $\theta$.

$$\begin{aligned} P(z|x) &= \text{softmax}(h_\theta(z, x)) \\ &= \frac{\exp(h_\theta(z, x))}{\sum_{z'} \exp(h_\theta(z', x))} \end{aligned} \quad (1)$$

Different from neural networks, human-like reasoning belongs to the deterministic type based on the DAG. This reasoning process should be probabilized to match the reasoning process of LMs. The Bayesian network is to build the

relationship among probabilities based on DAG. Therefore, it is suitable to probabilize the human reasoning process by converting the DAG to a probability graph, making it possible to parse LMs. According to the topological structure of the probability graph, the conditional probability distributions (CPD) of a set of random variables $(x_1, x_2, x_3, ..., x_n)$ can be investigated. Therefore, we can use the Bayesian network based on the DAG to measure LMs.

### 3.2 Probabilistic Probability Analysis

Based on intermediate results, we can perform the analysis by comparing prior probabilities from the Bayesian network and PTLMs. PTLM's probability $(P)$ can be computed by PTLMs directly:

$$P = P(\alpha|\psi) \tag{2}$$

where $\alpha \in \{\alpha_1, \alpha_2, \phi\}$, $\alpha_1, \alpha_2$, and $\phi$ are intermediate results and the hypothesis, and $\psi$ is the context splicing all premises.

To calculate the Bayesian probabilities of intermediate results, we should first define the initial probabilities. For the example in Figure 1, we can take advantage of PTLMs to calculate the conditional probabilities of $A$, $B$, $C$, $D$, $F$, and their negation propositions conditional on the context $\psi$. These probabilities are used as the initial probabilities, which can be represented by $P(\psi_i|\psi)$ and $P(\bar{\psi_i}|\psi)$, where $\psi_i \in \{a, b, c, d, f\}$, where $a \in \{A, \bar{A}\}$, and $b$, $c$, $d$, and $f$ are similar. Actually, we regard the process to reason out $\psi_i$ as a perceptual process because $\psi_i$ can be judged directly by the context $\psi$ without the reasoning process. As the whole reasoning process should be based on the perceptual process, it is reasonable to take these probabilities as the bases of Bayesian probabilities.

Based on initial probabilities, we consider the probability of the intermediate result $\alpha_1$ as a factorized representation of the distribution, and it is computed by the product of probabilities of correct premises ($A$ and $B$) that are localized probabilities. Similarly, conditional probabilities of each hop can be calculated iteratively. We define $P^*$ to represent the value calculated from the probability distribution.

$$P^*(\alpha_1|A, \psi) = \sum_{b \in \{B, \bar{B}\}} P(\alpha_1|A, b, \psi) P(b|A, \psi) \tag{3}$$

From the probabilistic DAG, different premises, $A$ and $B$, are independent conditional on $\psi$ ($A \perp B|\psi$). Therefore, Equation 3 can be rewritten as Equation 4:

$$P^*(\alpha_1|A, \psi) = \sum_{b \in \{B, \bar{B}\}} P(\alpha_1|A, b, \psi) P(b|\psi) \tag{4}$$

Another premise $B$ is similar to $A$, so the Bayesian probability of the intermediate result $\alpha_1$ can be calculated. Also, the calculation can be further simplified by the independent condition of ($\alpha_1 \perp \psi|a, b$), shown in Equation 5:

$$\begin{aligned} P^*(\alpha_1|\psi) &= \sum_{a \in \{A, \bar{A}\}} P^*(\alpha_1|a, \psi) P(a|\psi) \\ &= \sum_{\substack{a \in \{A, \bar{A}\} \\ b \in \{B, \bar{B}\}}} P^*(\alpha_1|a, b, \psi) P(b|\psi) P(a|\psi) \\ &= \sum_{\substack{a \in \{A, \bar{A}\} \\ b \in \{B, \bar{B}\}}} P^*(\alpha_1|a, b) P(b|\psi) P(a|\psi) \end{aligned} \tag{5}$$

---

(Input Premises)
F: If someone sees the rabbit then they like the rabbit.
A: The bear needs the tiger.
B: If someone needs the tiger then the tiger sees the cat.
C: If the tiger sees the cat then the cat chases the bear.
D: If someone chases the bear then they need the lion.
*Hypothesis*(Node-3): The cat needs the lion.
*Answer*: True
*Reasoning Path*: A + B $\longrightarrow$ C $\longrightarrow$ D
(Intermediate results)
$\boldsymbol{\alpha_1}$(Node-1): The tiger sees the cat.
$\boldsymbol{\alpha_2}$(Node-2): The cat chases the bear.
(Negation Examples)
$\bar{A}$: The bear does not need the tiger.
$\bar{B}$: If someone needs the tiger then the tiger does not see the cat. (CWA)
$\bar{\alpha_1}$: The tiger does not see the cat.
$\bar{\alpha_2}$: The cat does not chase the bear.

Figure 2: An example of a multi-hop instance, including premises ($\psi$), hypothesis ($\phi$), label, reasoning path, intermediate results ($\alpha_1, \alpha_2$) and some negation examples of premises.

Based on the same calculations, all prior probabilities of intermediate results and the final hypothesis ($P^*$) can be obtained from the theoretical Bayesian network and PTLMs. We utilize the Kullback-Leibler (KL) divergence to measure the difference of their distributions, which is used to analyze the PTLMs' reasoning ability. The calculation follows Equation 6, where $x \in \{\alpha_1|\psi, \alpha_2|\psi, \phi|\psi\}$.

$$KL(P||P^*)) = \sum_{i=1}^{N} (P(x_i) \log \frac{P(x_i)}{P^*(x_i)}) \tag{6}$$

Where $P^*$ represents the probability result based on Bayesian calculation and $P$ is the direct PTLMs' inference result.

## 4 Experiment

### 4.1 Experimental Settings

**Dataset** We perform analysis on the Soft Reasoner dataset proposed by Clark et al. [Clark *et al.*, 2020] and modify its multi-hop sub-sets by introducing the intermediate results as nodes and their negative propositions based on close-world assumption (CWA), shown in Figure 2. Based on the method of dataset construction, we constructed the intermediate results (such as $\alpha_1$ and $\alpha_2$) of multi-hop reasoning in a semi-automated way.

**Pre-trained Language Model** We select two fundamental PTLMs (BERT-large and RoBERTa-large), hyper-parameters of which are the same to make fair comparisons. We have trained each model more than three times and the hyper-parameters for training are shown in the Tabel 1. The target of PTLMs is to predict true/false for each hypothesis (or intermediate results) conditional on premises.

### 4.2 Result and Analysis

We first fine-tune PTLMs on the 2-hop training set and then evaluate them on the modified 3-hop dataset. Based on the

| Parameter | BERT | RoBERTa |
|---|---|---|
| Emb. Dim. | 1024 | 1024 |
| Max Length | 256 | 256 |
| LR | $5e^{-5}$ | $5e^{-5}$ |
| $LR_2$ | $5e^{-6}$ | $5e^{-6}$ |
| L2 | $1e^{-7}$ | $1e^{-7}$ |
| LR Decay | 1.0 | 1.0 |
| Epochs | 30 | 30 |
| Early Stop | 4 | 4 |
| Optimizer | ADAM | ADAM |

Table 1: Hyper-parameters for all models. Emb. Dim. is the dimension of embeddings. LR represents learning rate on the linear layer, while $LR_2$ represents learning rate on the PTLMs. L2 represents L2 regularity. ES means early stop.

Bayesian network, KL divergence of two intermediate results (KL-1 and KL-2) and the hypothesis (KL-3) are calculated. In this setting, the KL-1 and KL-2 are two in-domain metrics because the maximum reasoning hop of intermediate results is exactly 2, while KL-3 is an out-of-domain metric. Next, we evaluate BERT and RoBERTa on an in-domain test set (2-hop) and three out-of-domain test sets (3-hop, 5-hop, and zero-shot) from the original Soft Reasoner dataset. These out-of-domain, to a extent, can characterize the generalization ability of the trained model. These evaluations take the accuracy as the metric. Results are shown in Table 2.

| Domain | Metric | BERT | RoBERTa |
|---|---|---|---|
| **In** | 2-hop (%) | 98.2 | 99.2 |
| | KL-1 | 5.74 | 0.16 |
| | KL-2 | 7.70 | 0.28 |
| **Out** | 3-hop (%) | 83.5 | 91.2 |
| | 5-hop (%) | 56.7 | 79.3 |
| | zero-shot (%) | 85.7 | 93.1 |
| | KL-3 | 10.11 | 1.51 |

Table 2: Results of Bayesian analysis of BERT and RoBERTa. KL means the KL divergence between Bayesian probability and PTLM's probability. Other metrics are accuracies on the corresponding test set.Kl-1 and KL-2 are the KL scores over 1-hop and 2-hop examples respectively.

From Table 2, there is no significant difference about the accuracy of the model's judgment on the final result between BERT and RoBERTa evaluated on the in-domain set. However, RoBERTa has significantly lower in-domain KL metrics (0.16 of KL-1 and 0.28 of KL-2) than BERT (whose KL metrics are 5.74 and 7.70, respectively), which means that RoBERTa's reasoning process is more in line with the Bayesian reasoning process. This result is evidence that even if a model can make correct predictions, its prediction process does not necessarily conform to the human reasoning process.

Considering out-of-domain evaluations related to generalization, RoBERTa performs surprisingly better than BERT on all three test sets, which means that RoBERTa has better generalization ability in both more-hop reasoning (3-hop and 5-hop) and unseen (zero-shot) scenarios. Note that these results

are consistent with in-domain KL metrics, which is evidence that smaller KL metrics reflect better generalization to more complex scenarios. This conclusion conforms to the discovery of Wang et al. [Wang *et al.*, 2019].

In general, experimental results support the intuition that if a model can make probabilistic reasoning like humans, it will have better generalization ability. We can conclude that RoBERTa is more powerful to understand logical rules and apply them to reason than BERT, which conforms to the work of Talmor et al. [Talmor *et al.*, 2020]. In this sense, our analytical method provides a practical way to initially compare the generalization abilities of different neural models through KL metrics even without out-of-domain evaluation datasets.

### 4.3 Case Study
We perform a case study of the case in Figure 2. Its Bayesian probabilities and PTLMs' probabilities are displayed in Table 3.

| Propositions | BERT | | RoBERTa | |
|---|---|---|---|---|
| | Bayesian | Model | Bayesian | Model |
| Node-1($\alpha_1$) | 0.19 | 1.00 | 0.98 | 1.00 |
| Node-2($\alpha_2$) | 0.13 | 0.54 | 0.96 | 1.00 |
| Node-3($\phi$) | 0.00 | 1.00 | 0.86 | 1.00 |

Table 3: Case study by comparing Bayesian probabilities and PTLMs' probabilities.

From Table 3, RoBERTa can make the correct prediction with the probability of 1.00, roughly consistent with the Bayesian probability of 0.86. Although BERT can make the correct prediction of the intermediate results and the final hypothesis $\phi$ with the probability of 1.00, its Bayesian reasoning process gives the opposite conclusion with the probability of 0.00. Considering the accuracy, although such reasoning process of BERT does not conform to the human reasoning process, it is still regarded as successful reasoning. However, the KL metric considers the difference between BERT's probabilities and Bayesian's probabilities, allowing it to reflect such a spurious condition. Therefore, the KL metric can describe the generalization ability of PTLMs even if no out-of-domain evaluation is performed, but the in-domain accuracy cannot.

## 5 Conclusion
Although pre-trained language models (PTLMs), such as BERT and RoBERTa, have achieved great success in many NLU tasks, it is still challenging to understanding their true reasoning ability in the multi-hop reasoning scenarios. In this work, we propose a novel probabilistic analytical method to explore PTLMs' reasoning ability based on the constructed reasoning process (intermediate results). Specifically, we simulate the reasoning process as a Bayesian network that is a human-like reasoning process. Experiments on logical reasoning datasets, Softer Reasoner, provides a new view that human-like neural models (fitting the Bayesian network) have a better ability to generalize. Similarly, it provides thoughts for adding the Bayesian probability process to neural network analysis in the future.

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
