# OpenReview forum: "Exploring Multi-hop Reasoning Process in NLU from the View of  Bayesian Probability"
_ijcai.org/IJCAI/2021/Workshop/NSNLI — NSNLI Oral_

### Official Review · Reviewer_ZjQz · 2021-05-21
**Review of ‚Multi-hop Reasoning Analysis Based on The Bayesian Probability‘**

**Rating:** 6
**Confidence:** 3

**Review:**

The paper ‚Multi-hop Reasoning Analysis Based on The Bayesian Probability‘ investigates whether the estimated reasoning process of a pretrained transformer (PTLM) is consistent with the probabilistic reasoning process of a Bayesian network.
For this, the authors perform experiments on the ‚Soft Reasoner‘ multi-hop data set.
They predict the probabilities of intermediate results with the PTLMs and compare them to the probabilities of the intermediate results directly computed from the predicted probabilities of the hypotheses.
They find that among two flavours of PTLMs, the one that produces probabilities that are closer to the ones estimated with the Bayesian network generalises better.
As such, the work is relevant to the workshop.

A few things were unclear to me:
- What is the relation of this paper to work on calibration of neural networks, as a miscalibration of the PTLM would certainly influence the reported results?
- What is the accuracy for intermediate propositions? Does it differ between the models?
- Are the PTLMs trained only on predicting the truth value of the hypothesis or also to predict the truth value of the intermediate propositions?
- What is Emb. Dim. in Table 1? If it is the embedding dimension of the PTLMs, why isn’t it 1024?
- What is the early stopping criterion and what is the ‚4‘ given for ES in Table 1?


Typos:
- The title seems to be broken off

---

### Official Review · Reviewer_A1Ro · 2021-05-24

**Rating:** 6
**Confidence:** 3

**Review:**

This paper analyses whether reasoning capabilities of a pretrained language model can be compared to a Bayesian network for a probabilistic reasoning process.
An interesting apporach of the paper is to test this out via intermediate results in the k-hop reasoning environment, where the model's probability of predicting an intermediate result 'agrees' with a probabilistic reasoning model that 'simulates'/aligns with human reasoning.
The results section shows that RoBERTa model performs better than the BERT model, i.e. aligns with a probabilistic reasoning model better.
This work seems like an interesting approach to analysing reasoning capabilities of language models and might spark further interest in this area.


I found the following two things unclear:
- "the analytics method should be capable of measuring the probability distributions of the whole reasoning process rather than only provide deterministic reasoning results" - I understood this as the language models should be able to measure the probability distributions of reasoning, which is not the case given that their probabilistic approach to language modeling is not trained specifically for reasoning. The reasoning elements these models learn seems to be more of a side-effect than the goal, in the case where they're trained as language model. In this paper they are trained on a specific reasoning dataset so that differs. I would suggest clarifying this sentence.
- "Neural networks make reasoning through a fully-connected bidirectional graph" - On a high-level this seems like a plausible connection, but the letter choice f, a, b, c, d and F, A, B, C, D seems to imply a conenction there which is difficult to establish given that language models learn premises token-by-token or word-by-word, and not premise-by-premise. Please clarify this and make the connection more explicit (an example would be great).




Smaller issues:
- the title is cut off
- "due to the probabilistic characteristics of PTLMs [Manhaeve et al. 2018]"; the work of Manhaeve et al does not consider language models so it seems like an inappropriate citation to support the probabilistic nature of PTLMs
- Figure 1 mentions red-bordered circles, but there are no red bordered circles. Also, premises \psi are mentioned, but \psi is nowhere to be found in the figure
- what are [CLS] and [SEP], please define
- close world -> closed-world
- trained each model three times - what does this mean? Trained with a different random seed? Why three?

---

### Decision · Program_Chairs · 2021-05-27

**Decision:**

Accept (Oral)

**Comment:**

The paper tackles the topic clearly relevant for the workshop.
Please take the reviewers' comments into account when preparing the camera-ready version.